# OpenReview forum: "Backdooring CLIP through Concept Confusion"
_ICLR.cc/2026/Conference — ICLR 2026 Conference Withdrawn Submission_

### Official Review · Reviewer_rZ4E · 2025-10-30

**Soundness:** 3
**Presentation:** 2
**Contribution:** 2
**Rating:** 2
**Confidence:** 4

**Summary:**

This paper proposes a backdoor attack called Concept Confusion Attack (C²Attack), which uses human-understandable concepts learned internally by the model as triggers, rather than traditional external pixel patterns, thereby improving the stealth of the backdoor model and its ability to evade existing defenses.

**Strengths:**

- Using CLIP’s concept representations as the internal backdoor trigger, which is stealthy.

- The effectiveness of the attack is verified on multiple datasets (CIFAR-10/100, Tiny-ImageNet) and multiple CLIP architectures.

- Experiments are conducted on various defense and attack methods.

**Weaknesses:**

- One of the main contributions is "employ internal concepts as triggers, eliminating the need for external patterns". But there are also other works that explored this idea [A, B, C].

- The paper only includes image classification. Why did you choose CLIP as your architecture? Are the concept-interpretation techniques compatible with other architectures, such as ResNet?

- While the attack method is effective, there is a lack of specific recommendations or preliminary defense experiments for the proposed attack.

- The details of training settings are not provided, which limits the reproducibility.

- Writing can be more concrete. For example, explain the definition of "concept" when the word first appears.

**Questions:**

Thank you for the interesting work, but I still have a few concerns.

- This paper attributes one of its contributions to "exploring internal concepts as triggering factors". But this has been explored by existing works from two perspectives. Firstly, [A] explores the natural backdoors. Without changing the training dataset at all, adversaries can exploit internal patterns as a natural backdoor. This is even better than the C^2 attack, because it does not change the label. Therefore, I suggest that the authors should discuss more about the difference between the C^2 attack and existing works.

- Summary of traditional backdoor attacks (Figure 1) ignores many better backdoor attacks, such as WaNet, Lira, Bpp, Clean label backdoor, etc. Can you explain why the proposed internal backdoor (C^2 attack) can be more stealthy than external invisible triggers? And why do we want this "no changes on input"?


[A] Rethinking Backdoor Attacks

[B] Narcissus: A Practical Clean-Label Backdoor Attack with Limited Information

[C] Towards Backdoor Stealthiness in Model Parameter Space

---

### Official Review · Reviewer_rt8D · 2025-10-31

**Soundness:** 2
**Presentation:** 3
**Contribution:** 2
**Rating:** 4
**Confidence:** 4

**Summary:**

The paper proposes a poisoning attack for CLIP-based image classifiers in which images that strongly exhibit a concept (chosen via concept extractor) are relabeled to a target class, such that the natural presence of the concept would function as a trigger at test time. The authors report high ASR with competitive clean accuracy on CIFAR-10/100 and Tiny-ImageNet, and present comparisons against several attacks and defenses.

**Strengths:**

• The paper presents an intuitive and simple attack mechanism using label-only manipulation (no input patch) that leverages a model’s internal concepts as triggers, which in turn makes it robust against multiple defenses.
• The authors highlight the limitations of current defenses that rely on input-level modifications to flag anomalous behavior.
• Extensive experiments across multiple datasets, encoder scales, concept trigger choices, and ablations under the proposed settings demonstrate the effectiveness of the proposed attack procedure.

**Weaknesses:**

• The paper’s primary claim is that it has created a "concept-level" backdoor. However, it’s unclear whether model has learned to associate the general, abstract concept of e.g., "water" with the target label. This is especially important when concept=target class, which can reduce to reweighting unless target-class images are excluded from poisoning and ASR is measured on concept-present but non-target images.
• The paper investigates defenses that primarily rely on input level anomalies; while compelling for their purpose, a proper comparison is needed against post-hoc clean data fine tuning defenses. The closest defense discussed is Fine-Pruning, which the paper consistently refers to as fine- tuning in their tables and neither fully state implementation details (e.g., dataset size, training hyperparameters) nor does it evaluate against its simpler variant (e.g., vanilla CE finetuning on clean data).
• A comparison to Finding Naturally Occurring Physical Backdoors in Image Datasets (NeurIPS’22) is needed to contextualize the contribution, given the similar “semantic trigger via relabeling” premise.
• Since the main focus of the paper is on CLIP – and the authors compare against image-caption optimized attacks such as BadCLIP – it is important to have at least a single experiment with image-caption data.
• Multiple LLM usage artifacts "CopyRetryClaude can make mistakes. Please double-check re-sponses." appear in App. B.3 and App. B.4.

**Questions:**

• Does a trained concept trigger transfer out of distribution (e.g., CIFAR-10 backdoored model evaluated with ImageNet-1K images containing the concept)? Can the concept be successfully superimposed on image to cause attack (e.g., adding an ’Airplane’ on clean images)?
• What are the different targets for each concept trigger evaluated in Table 2?
• What does using ’Airplane’ as concept and target label mean? How is the ASR evaluated in this case? (See lines 460-461, 467-468 and 473-474)
• How scalable is the full-dataset concept scoring on larger datasets (e.g., ImageNet-1k or CC3M), can the attack be realized using a subset of the dataset?
• Why was simple finetuning (on entire model) with clean data not evaluated as a baseline?
• How were the physical baselines on CIFAR-10 (Table 5) implemented (trigger choice, poison rate, training details, etc.)?

---

### Official Review · Reviewer_BPdZ · 2025-11-01

**Soundness:** 1
**Presentation:** 2
**Contribution:** 1
**Rating:** 2
**Confidence:** 4

**Summary:**

The paper proposes the Concept Confusion Attack (C^2 Attack), a new backdoor attack against the CLIP encoder. The C^2 Attack moves the attack from classical input-space triggers, such as adversarial perturbations or patches, to human-interpretable concepts present in the image. In particular, if an image exhibits a concept chosen by the attacker, the model is finetuned to misclassify it to a target class. The authors show the attack achieves high ASR, maintains clean accuracy, and evades several tested defenses.

**Strengths:**

- Identifying backdoor vulnerabilities in vision foundation models is a timely and relevant problem.
- To the best of my knowledge, the proposed attack vector is novel.
- The paper is generally well-written and easy to follow.

**Weaknesses:**

- Unlike traditional attacks, the attacker has no active control over the attack, which, in my opinion, makes the threat model impractical, if not useless. The attacker must passively wait for an image that 'naturally' contains the attacked concept (above an arbitrary threshold) and cannot trigger the attack on an arbitrary input. The very concept of a "clean image" also becomes ill-defined in this framework.
- The attack, as presented, strongly couples the concept threshold $\sigma$ to the poisoning ratio.
- There is a contradiction regarding the poisoning ratio. Section 4.2 claims a 1% rate, while Appendix B.5 states 99%. A 99% rate is not a backdoor attack: it is a full dataset retraining on corrupted data, which would invalidate all experimental claims of stealth and effectiveness.
- All modern, state-of-the-art defenses for CLIP (CleanCLIP, RoCLIP, etc.) are missing from the evaluation. The baselines that are evaluated are weak and outdated.
- The attack success rate significantly decreases with data complexity (even at the Tiny ImageNet scale). The reliance on CIFAR-scale results is insufficient. More practical, large-scale benchmarks such as ImageNet or CC3M are missing.
- The paper repeatedly affirms that previous attacks rely on "visible patterns", which is factually incorrect. This ignores a large body of work on imperceptible attacks.

**Questions:**

- Table 2: Can the authors quantify the prevalence of the trigger concepts in CIFAR-10? For example, how many images actually contain a "clock", "air conditioner", or "cushion"? The analysis in Section 4 appears overly fine-grained for $32 \times 32$ images that typically contain centered, single objects.
- Section 4.1: Is it not expected that this shift would be more pronounced in deeper layers, as the change must propagate to the output to affect the final decision? Moreover, the statement that a backdoor "can be interpreted as manipulation of concepts" seems trivial. What is the null hypothesis this analysis is intended to disprove?

---

### Official Review · Reviewer_bqqe · 2025-11-11

**Soundness:** 2
**Presentation:** 2
**Contribution:** 2
**Rating:** 4
**Confidence:** 3

**Summary:**

The paper proposes a concept-level backdoor attack on CLIP that manipulates internal representations instead of adding visible triggers. It relabels images containing a chosen concept (e.g., “water”) to a target class, causing the concept itself to trigger misclassification. Experiments on CIFAR and Tiny-ImageNet show high attack success and robustness against standard defenses, revealing a new vulnerability in multimodal models.

**Strengths:**

This paper is well-presented and clearly written, with a logical structure and comprehensive experimental evaluation. By shifting the attack surface from pixel-space triggers to semantic concepts, the work highlights a new and underexplored vulnerability in multimodal foundation models. The experiments are thorough and demonstrate the feasibility and effectiveness of the approach.

**Weaknesses:**

Practical threat scenario. While the proposed Concept Confusion Attack is novel, the paper does not clearly define a realistic threat scenario compared to other backdoor attacks. It assumes full control over the training data and process, which is a strong assumption. In practice, this limits the relevance of the attack since most CLIP or foundation models are pre-trained on large-scale, heterogeneous data that attackers rarely control. The paper would benefit from discussing more realistic scenarios where concept-level manipulation could plausibly occur.

Lack of adaptive defense discussion. The defense evaluation section focuses on existing input- and representation-based defenses, but it does not explore adaptive defenses that could exploit model-behavioral changes induced by concept manipulation. Since the attack directly alters internal concept–label associations, it likely introduces detectable shifts in latent representations or output distributions. These changes could make the model distinguishable from a clean one using concept-consistency checks or embedding-space monitoring. A discussion or experiment on such adaptive detection would strengthen the claim of stealthiness.

Dirty-label nature of the attack. The proposed attack relies on relabeling samples that contain the target concept (e.g., labeling “duck on water” as “boat”), which qualifies as a dirty-label attack rather than a clean-label one. This distinction should be made explicit. The paper should clarify this limitation and justify why this label manipulation is realistic or necessary for the proposed method.

**Questions:**

Please discuss the threat scenario, adaptive defense especially regarding the distribution shift, and dirty-label setup.

---

### Note · Authors · 2025-11-13

I have read and agree with the venue's withdrawal policy on behalf of myself and my co-authors.